# The Effect of Chiropractic Treatment on Limb Lameness and Concurrent Axial Skeleton Pain and Dysfunction in Horses

**DOI:** 10.3390/ani12202845

**Published:** 2022-10-19

**Authors:** Mikaela D. Maldonado, Samantha D. Parkinson, Melinda R. Story, Kevin K. Haussler

**Affiliations:** 1Equine Orthopaedic Research Center, Department of Clinical Sciences, College of Veterinary Medicine and Biomedical Sciences, Colorado State University, Fort Collins, CO 80523, USA; 2Department of Veterinary Preventative Medicine, The Ohio State University, Columbus, OH 43210, USA

**Keywords:** manual therapy, back pain, stiffness, lameness, hypertonicity, mechanical nociceptive thresholds, inertial sensor

## Abstract

**Simple Summary:**

The use of chiropractic techniques is common in horses and a strong body of evidence exists for effectively treating back pain and stiffness. Chronic limb lameness can induce complex interactions with the neck, back and pelvis in affected horses, which can be a challenging clinical issue with limited available conservative treatment options. We used a comprehensive array of tests to measure lameness, pain, stiffness, and muscle hypertonicity to evaluate the global effects of chiropractic care in horses with chronic lameness. Four chiropractic treatment sessions were applied over 3 weeks. Improvements in subjective measures of lameness, back muscle pain, and neck or back stiffness were noted. Further studies are needed to better identify the type and severity of lameness that may be amendable to chiropractic treatment.

**Abstract:**

Chiropractic care is a common treatment modality used in equine practice to manage back pain and stiffness but has limited evidence for treating lameness. The objective of this blinded, controlled clinical trial was to evaluate the effect of chiropractic treatment on chronic lameness and concurrent axial skeleton pain and dysfunction. Two groups of horses with multiple limb lameness (polo) or isolated hind limb lameness (Quarter Horses) were enrolled. Outcome measures included subjective and objective measures of lameness, spinal pain and stiffness, epaxial muscle hypertonicity, and mechanical nociceptive thresholds collected on days 0, 14, and 28. Chiropractic treatment was applied on days 0, 7, 14, and 21. No treatment was applied to control horses. Data was analyzed by a mixed model fit separately for each response variable (*p* < 0.05) and was examined within each group of horses individually. Significant treatment effects were noted in subjective measures of hind limb and whole-body lameness scores and vertebral stiffness. Limited or inconsistent therapeutic effects were noted in objective lameness scores and other measures of axial skeleton pain and dysfunction. The lack of pathoanatomical diagnoses, multilimb lameness, and lack of validated outcome measures likely had negative impacts on the results.

## 1. Introduction

Lameness is a common cause of poor performance in horses [1]. The clinical objective in managing horses with acute lameness is to localize the site of nociception via clinical examination and the use of diagnostic local anesthesia (i.e., where is the source of pain). Diagnostic imaging is then used to help identify the potential tissue injury that is contributing to the clinical signs (i.e., what is the source of pain). However, lameness may not always be readily localized with local anesthesia due to variable diffusion or poor technique [2,3]. Similarly, diagnostic imaging may not always provide a definitive diagnosis despite the localization of lameness with diagnostic anesthesia [4,5]. Therefore, the localization of the source of pain that contributes to signs of lameness is not always clear and may not be limited to a single site or within a single limb [6,7].

Pain originating within a limb often induces altered limb loading and abnormal movement patterns with the potential for producing overuse injuries within the axial skeleton [8]. In clinical cases that do not have clearly localizable limb lameness, subtle neurological disorders (e.g., weakness, lack of impulsion), back pain and stiffness, or sacroiliac dysfunction have been judged to be significant contributing factors to the observed altered gait patterns [6,9,10]. Specifically, back pain can be both a cause and a result of lameness [11]. Reports suggest that 23–32% of horses with limb lameness may have concurrent back pain and 68–85% of horses with primary back pain may have a concurrent limb lameness [12,13,14,15]. There is increasing evidence of the clinical importance of compensatory mechanisms and interactions between the axial and appendicular regions with regard to pain and lameness [16].

Limb lameness is commonly treated through the administration of nonsteroidal anti-inflammatory drugs (NSAIDs), corticosteroids, or biological therapies [17,18,19]. However, limitations exist in their administration due to potential adverse effects with long-term use, expense, and competition restrictions that limit performance enhancing substances [20,21]. The use of NSAIDs or corticosteroid injections may not always be effective or appropriate for managing chronic back pain or compensatory lameness issues [22,23]. Effective management strategies for chronic pain and lameness often require a multimodal approach that includes local, regional, and systemic treatment [24].

Most sport horse practitioners have access to a wide array of therapeutic modalities, which often include nonpharmaceutical approaches for managing chronic lameness and axial skeleton disorders [25,26]. Chiropractic is commonly used in equine practice as an adjunctive treatment for managing chronic back pain and lameness [27,28]. A systematic review of spinal manipulation techniques suggests a high level of efficacy in reducing thoracolumbar pain, stiffness, and muscle hypertonicity in horses [29]. While chiropractic care has a strong body of evidence for treating equine back pain and dysfunction, there is limited evidence for its effect on the appendicular skeleton and associated limb lameness [30]. Chiropractic has been reported to induce minor kinematic effects in limb movements within sound horses treated for back pain [31]. The objective of this study was to evaluate the efficacy of chiropractic treatment in reducing the clinical signs of chronic limb lameness and concurrent axial skeleton pain and dysfunction in horses. We hypothesized that chiropractic treatment would affect global measures of limb lameness and axial skeleton function, irrespective of the perceived sites or sources of pain.

## 2. Materials and Methods

### 2.1. Subjects

Twenty mixed-breed horses used in a collegiate polo program were enrolled into the study based on the criteria of having a grade 1–3/5 lameness (American Association of Equine Practitioners (AAEP) scale) within at least one fore or hind limb. The polo horses were enrolled at the beginning of their competition season. To reduce the variability in outcome parameters observed in the polo horses, a second group of horses were enrolled in year 2, which included eighteen privately owned Quarter Horses that were active in ridden or competitive work with a primary hind limb lameness of grade 1–3/5 localized to within at least one hind limb.

### 2.2. Inclusion-Exclusion Criteria

In year 1, all polo horses were subjectively evaluated for the presence of fore and hind limb lameness and horses with lameness scores > 3/5 were excluded from the study. In year 2, Quarter Horses were selected for the presence of hind limb lameness and were excluded if they had a primary forelimb lameness or a positive response to distal hind limb flexion tests that was indicative of lower limb lameness. All owners provided informed consent prior to inclusion in the study and were asked to refrain from providing additional supplements, medications, or ridden exercise outside of their typical routine throughout the study duration to limit confounding factors. Owners were blinded to the assigned treatment group.

### 2.3. Study Design

The study was a blinded, randomized, controlled clinical trial that included two separate groups of horses that were involved in different disciplines (polo, Western performance) for which data was collected over two different time points (year 1, year 2). Horses were numerically randomized to treatment and control groups. Treatment consisted of whole-body chiropractic evaluation and treatment, while the control group received no active treatment. A single blinded observer (S.D.P. in year 1; M.D.M. in year 2) performed all spinal evaluation procedures prior to any applied treatment across years. All evaluations were performed with the horses standing quietly on firm ground in a familiar environment to reduce stress and limit variability. Outcome parameters were recorded on days 0, 14, and 28, which included subjective and objective lameness evaluation, detailed spinal examination, assessing active spinal range of motion, induced spinal reflexes, and mechanical nociceptive thresholds.

### 2.4. Subjective Lameness Evaluation

Subjective lameness evaluations within each group of horses were conducted by a single blinded and experienced observer (S.D.P. in year 1; M.R.S. in year 2). Horses were trotted over hard ground in straight lines and circles in both directions with lameness scored (0–5) for each individual limb. No attempts were made to identify a primary or secondary limb lameness in horses with multiple limb lameness. Lameness scores were summed across left-right fore and hind limbs to provide paired forelimb and hind limb lameness scores. Lameness scores were then summed across all limbs within horse to provide a whole-body measure of limb lameness (e.g., grade 1 in the left forelimb; grade 2 in the right hind limb; a whole-body lameness score of 3).

### 2.5. Objective Lameness Evaluation

A body-mounted inertial sensor system (Equinosis Q, Lameness Locator, Equinosis, LLC, Columbia, MO, USA) was used to assess fore and hind limb lameness using previously described methodology [32,33]. Each horse was trotted in a straight line on a firm surface with the goal of acquiring > 25 strides to provide sufficient data to be regarded as a reliable measure of lameness. A successful trial was defined as having a reported standard deviation of head-height differences less than ±6.0 mm for the forelimbs and pelvic-height differences less than ±3.0 mm for the hind limbs [34]. The presence and severity of forelimb lameness was based on reported values for maximum and minimum head-height differences (*HD_max_* and *HD_min_*, respectively). The total head-height difference was reported as a vector sum for the forelimbs, which is described as a global measure of forelimb lameness if the absolute value of the vector sum was >8.5 mm [33,34,35]. Hind limb lameness was similarly quantified using maximum and minimum pelvic-height differences (*PD_max_* and *PD_min_*, respectively). Total pelvic-height difference was calculated as the absolute value of the vector sum for the hind limbs by taking the square root of the summed squares of the *PD_min_* and *PD_max_* values, as previously described [34]. A whole-body lameness score was calculated from the combined vector sum values of the fore and hind limbs using the following formula:$$Overall\;Vector\;Sum=\left(\left|Forelimb\;Vector\;Sum\right|-8.5\right)+\left[\frac{\left(\left|Pmax_{mean}\right|+\left|Pmin_{mean}\right|\right)}{3}\right]$$

### 2.6. Spinal Evaluation

A detailed spinal examination was performed using digital palpation to assess the location and perceived severity of pain, stiffness, and epaxial muscle hypertonicity within the head, cervical, thoracolumbar, and pelvic regions [36,37]. The thoracolumbar fascia and epaxial musculature were palpated with graded digital pressure to identify and localize sites of myofascial pain and muscle hypertonicity. Firm digital pressure was applied along the dorsal midline over the T4-S5 spinous processes to assess sensitivity. Stiffness was assessed using low amplitude, laterally directed oscillations applied segmentally over each intervertebral articulation from the occiput to sacrum [38]. A score for the judged severity of pain, stiffness, and epaxial muscle hypertonicity was assigned to each individual intervertebral segment on both the left and right sides of the axial skeleton using the following criteria: 0 = no abnormalities noted, 1 = mild, 2 = moderate, 3 = severe, and 4 = unable to evaluate. Pain, stiffness, and epaxial muscle hypertonicity scores were then summed within specified vertebral regions: cervical (occiput-C7), cranial thoracic (T3–T10), caudal thoracic (T11–T18), and lumbopelvic (L1–L6, pelvis, sacrum).

Whole-body scores of pain, stiffness, and epaxial muscle hypertonicity were also recorded based on the overall perceived severity and number of affected vertebral regions within individual horses. Severity was scored: 0 = no abnormalities noted, 1 = mild, 2 = moderate, 3 = severe, and 4 = complete avoidance and evasion from the applied pressure. Affected vertebral regions were scored: 0 = no abnormalities noted, 1 = positive findings noted within a single vertebral region, 2 = positive findings noted within two vertebral regions, 3 = positive findings noted within three vertebral regions, and 4 = positive findings noted across all vertebral regions.

### 2.7. Active Range of Motion

Baited stretches were performed to assess the range of motion and fluidity of induced active movement of the axial skeleton [39,40]. Five specific movements were induced on both the left and right sides within each horse, which included:Lateral bending of the cranial cervical region (Figure 1a): The treat was initially positioned approximately 12–18 inches lateral to the head and held at the height of the withers to maintain cervical extension. The treat was then moved caudally to direct the muzzle of the horse laterally and caudally until the horse’s head was facing caudally (i.e., perpendicular to the long axis of the trunk) or until the horse was no longer able to follow the treat (i.e., end range of motion).Lateral bending of the middle cervical region (Figure 1b): The treat was initially positioned approximately 12–18 inches lateral to the head and the horses’ muzzle was directed laterally and caudally toward the point of the elbow at the girth region with the neck maintained in a neutral flexion-extension position. Attention was focused on the ability to laterally bend the middle cervical region.Lateral bending of the caudal cervical region (Figure 1c): The treat was initially positioned approximately 12–18 inches lateral to the head and the horses’ muzzle was directed laterally and ventrally toward the lateral surface of the ipsilateral carpus to induce concurrent cervical flexion. Attention was focused on the ability to laterally bend the caudal cervical region around the ipsilateral scapula and shoulder region.Combined lateral bending of the cervical and thoracolumbar regions (Figure 2a): The horse’s tail was grasped with the caudal hand and lateral tension was applied until quadriceps muscle activation of the ipsilateral hind limb was observed. Simultaneously, the treat was positioned approximately 24–36 inches lateral to the girth region and the horses’ muzzle was directed laterally and caudally toward the stifle region. Attention was focused on the ability of the horse to touch and maintain the muzzle position at the stifle region.Combined flexion and lateral bending of the cervical and thoracolumbar regions: The same procedure was repeated, except that the treat was directed toward the ipsilateral tarsal region to induce concurrent trunk flexion and lateral bending. Attention was focused on the ability to activate the internal abdominal oblique muscle (Figure 2b).

The range of motion and the fluidity of the induced movements were graded (0–4) for all baited stretches. The range of motion was graded as 0 = readily able to reach and touch the muzzle to all targeted sites, 1 = 25% reduction in the range of motion, 2 = 50% reduction, 3 = 75% reduction, and 4 = unable to perform the requested movement. The baited stretches that were performed to the level of the stifle (#4) and tarsus (#5) were also scored based on the measured distance between the muzzle and the respective landmark. The fluidity of the induced movements was graded as 0 = smooth, controlled motion and able to hold the end range of motion position for 2–3 s, 1 = smooth, controlled motion and able to hold the end range of motion position for <1 s, 2 = smooth, controlled motion and not able to hold the end range of motion position, 3 = jerky, uncontrolled motion and not able to hold the end range of motion position, and 4 = unable to perform the requested movement. The range of motion and fluidity scores of the baited stretches were summed across left-right sides within the cervical region (#1 thru #3 above) and thoracolumbar vertebral regions (#4 and #5 above).

### 2.8. Spinal Reflexes

Digital stimulation was applied along the ventral midline over the sternum or cranial portion of the linea alba to induce elevation of the cranial thoracic region [38]. Bilateral digital stimulation at the lateral tail head was also used to induce a combined reflex of pelvic flexion and trunk elevation (i.e., kyphosis). Each spinal reflex was graded based on the fluidity and the ability to hold the induced movement and the amplitude of the induced movement. The range of motion and fluidity was graded (0–4) using the same criteria as reported above for the baited stretches. The response to firm bilateral compression of the tubera sacralia was also scored based on the presence of a pain avoidance response and unilateral or bilateral unlocking of the stifles. The scoring system used for judging the response to tubera sacralia compression was 0 = no perceived pain response or mild local muscle contraction, 1 = mild avoidance reaction and moderate lumbosacral extension, 2 = moderate avoidance reaction and inconsistent unlocking of one stifle, 3 = severe avoidance reaction and consistent unlocking of both stifles, and 4 = compete avoidance and evasion from the applied pressure. The range of motion and fluidity scores of the three spinal reflexes (i.e., sternal, pelvic, tuber sacrale) were summed within horse.

### 2.9. Mechanical Nociceptive Thresholds

A pressure algometer (Model FPK 40, Wagner Instruments, Wagner Instruments, Greenwich, CT, USA) with a 1-cm^2^ cylindrical, rubber tip was used to measure mechanical nociceptive thresholds (MNTs) of the left and right epaxial musculature at selected vertebral levels within the cervical (C2, C3, C4, C5), thoracic (T3, T9, T18), and lumbosacral (L3, L6, S2) regions. Pressure was applied perpendicularly at approximately 1 kg/cm^2^/sec until a local avoidance reaction is noted (i.e., skin twitching, local muscular contractions, or stepping away) [41]. Measurements were repeated 3 times per site approximately 3–4 s apart. The MNT values were summed across left-right sides within the cervical, thoracic, and lumbosacral regions.

### 2.10. Chiropractic Treatment

Horses were numerically randomized to treatment and control groups. Treatment consisted of high-velocity, low-amplitude, manually applied thrusts to sites of perceived pain or stiffness with the axial and appendicular articulations [42,43]. Treatment was applied on days 0, 7, 14, and 21 by a single examiner (K.K.H., years 1 and 2). The control group received no treatment and were restrained quietly for 15 min to simulate the time required for chiropractic treatment.

### 2.11. Statistical Analysis

Data was judged to be normally distributed based on the visual inspection of diagnostic plots, therefore all results were reported as means and standard deviations. A mixed model was fit separately for each response variable which included fixed effects of group (treatment or control), week ((baseline, week 2 and week 4), and group * week interactions. Horse was included as a random effect to account for repeated measures. Tukey HSD comparison was performed to compare left-right differences and year 1-year 2 differences within the measured outcome parameters. For most variables there were no statistically significant left-right differences in the measured parameters, therefore left-right data was pooled. For most variables, statistically significant differences were noted between the polo and Quarter Horses and data was reported separately. For each response variable, statistical comparisons were reported within groups over the course of the study (baseline, week 2 and week 4) and between treatment and control groups at each time point. JMP software (SAS Institute, Cary, NC, USA) was used for all statistical analyses with significance set at *p* ≤ 0.05.

## 3. Results

### 3.1. Subjects

The collegiate polo horses with mixed fore and hind limb lameness included 13 mares and 7 geldings with an age distribution of 15.2 ± 3.5 (range 5 to 23) years. The Quarter Horses with primary hind limb lameness were 11.7 ± 5.8 (range 6 to 22) years of age and included 13 geldings and 5 mares.

### 3.2. Subjective Lameness Evaluation

No significant treatment effect was noted in the subjective evaluation of forelimb lameness for the polo (Figure 3) or Quarter Horses (Figure 4). Within the Quarter Horses, the control group had higher, but not significantly different summed forelimb lameness scores across weeks (Table A1). Subjective forelimb lameness scores were higher in the polo (*p* = 0.001), compared to the Quarter Horses.

Subjective hind limb lameness scores decreased significantly across weeks in both the treatment and control groups in the polo horses (Figure 3). Hind limb lameness scores also decreased across weeks within the treatment group in the Quarter Horses (Figure 4); however, the change was not significant (*p* = 0.078). In the Quarter Horses, there was a significant treatment group difference in subjective hind limb lameness scores at week 4 (Table A2, *p* = 0.012). Higher, but not significant (*p* = 0.946), hind limb lameness scores were noted in the Quarter Horses (range 2.7 to 5.0), compared to polo horses (range 1.1 to 2.7).

Whole-body subjective lameness scores were significantly different across weeks in both the treatment and control groups in the polo horses (Figure 3). In the Quarter Horses, the treatment group had significantly different whole-body lameness scores across weeks (Figure 4); however, the week 4 score did not differ significantly from the baseline value (Table A3, *p* = 0.238). Whole-body subjective lameness scores tended to be higher, but not significantly different (*p* = 0.069) in the polo horses, compared to the Quarter Horses.

### 3.3. Objective Lameness Evaluation

Within the polo horses, a primary forelimb lameness was noted in 16 horses (8 within each of the treatment and control groups) and a primary hind limb lameness was noted in 4 horses (2 within each group). All Quarter Horses were enrolled based on the presence of primary hind limb lameness, which was confirmed on the inertial sensor analysis in 16 of 18 horses. The other 2 horses (within the control group) were judged to have a primary forelimb lameness and secondary hind limb lameness based on the inertial sensor analysis. Overall, a measurable degree of forelimb lameness was reported in 4 of 10 treatment and 3 of 8 control horses in the Quarter Horse group.

No significant treatment effects were noted in the severity of objective measures of forelimb (Table A4), hind limb (Table A5), or whole-body (Table A6) lameness scores, which was likely due to the large variability in measured values (Figure 5 and Figure 6). Whole-body objective lameness scores were not significantly different (*p* = 0.521) between the polo and Quarter Horses.

### 3.4. Spinal Evaluation

Treatment produced a significant reduction in pain severity within the caudal thoracic region in the polo horses (Figure 7). Non-significant reductions in pain were noted within the cranial (*p* = 0.075) and caudal thoracic (*p* = 0.052) regions in the Quarter Horses (Figure 8). Of note are the significantly higher pain scores within the lumbopelvic region (*p* < 0.0001) in the polo horses (Table A7, range 3.8 to 8.9), compared to Quarter Horses (range 0.6 to 1.1). Although non-significant, pain scores were higher (*p* = 0.394) in the polo, compared to Quarter Horses.

Reduced, but not significant, cervical (*p* = 0.055) and lumbopelvic (0.091) stiffness was noted within the treatment group in the polo horses (Figure 9). Measures of axial skeleton stiffness decreased significantly in the Quarter Horses across all vertebral regions within the treatment group (Figure 10). In general, relatively higher stiffness scores were noted in the Quarter Horses, compared to polo horses (Table A8), across all vertebral regions (cervical, *p* < 0.0001; cranial thoracic, *p* = 0.467; caudal thoracic, *p* < 0.000; and lumbopelvic, *p* < 0.0001).

Treatment had no significant effect on measures of muscle hypertonicity across vertebral regions (Table A9). Cervical muscle hypertonicity scores were significantly higher (*p* = 0.264) than other vertebral regions in the polo horses (Figure 11). Control group scores of muscle hypertonicity were significantly increased within the caudal thoracic (*p* = 0.034) and lumbopelvic (*p* = 0.012) regions in the Quarter Horses (Figure 12). Muscle hypertonicity scores were noticeably higher across vertebral regions in the Quarter Horses, compared to polo horses (cervical, *p* = 0.499; cranial thoracic, *p* < 0.0001; caudal thoracic, *p* < 0.0001; and lumbopelvic, *p* < 0.0001).

Treatment produced significant reductions in whole-body scores of pain severity (*p* = 0.041) in the polo horses (Figure 13) and the number of affected vertebral regions with stiffness (*p* = 0.030) in the Quarter Horses (Figure 14). Whole-body scores of muscle hypertonicity severity significantly increased in the control group in the Quarter Horses (Figure 15, Table A10). Whole-body scores for combined pain, stiffness and muscle hypertonicity were significantly higher (*p* < 0.0001) in the Quarter Horses, compared to polo horses.

### 3.5. Active Range of Motion

Summed scores for active range of motion tended to decrease (i.e., improve) over time within the treatment groups, compared to the control groups; however, none of these changes were statistically significant (Figure 16). The active range of motion scores for the thoracolumbar region across both treatment and control groups were notably higher (i.e., less fluid or reduced range of motion; *p* < 0.0001) in the polo horses (range 7.4 to 8.8), compared to the Quarter Horses (range 2.7 to 5.6, Table A11). The measured distances from the muzzle to the stifle or tarsus were not significantly different across weeks or years (all *p* > 0.05).

### 3.6. Spinal Reflexes

No significant changes were noted for summed spinal reflex scores across weeks or years (Figure 17). Spinal reflex scores were significantly lower (i.e., better fluidity and range of motion; *p* = 0.013) in the Quarter Horses (range 1.8 to 3.4), compared with the polo horses (range 3.1 to 4.7). Spinal reflex scores tended to decrease (i.e., improve) within the control group of polo horses and treatment groups in both polo and Quarter Horses (Table A12); however, these changes were not statistically significant (*p* = 0.090 to *p* = 0.435).

### 3.7. Mechanical Nociceptive Thresholds

MNT values decreased (i.e., more painful) within the cervical region for both treatment and control groups in the polo horses (Figure 18). The lumbosacral MNT values significantly increased (i.e., less painful) within the treatment group in the Quarter Horses (Figure 19). MNT values within the cervical region were significantly higher in the polo horses (*p* < 0.0001), compared with the Quarter Horses (Table A13), with nonsignificant differences within the thoracic (*p* = 0.549) and lumbosacral regions (*p* = 0.608) across years.

## 4. Discussion

The objective of this study was to evaluate the effects of chiropractic treatment on global measures of limb lameness and concurrent axial skeleton pain and dysfunction. Overall, there were positive treatment effects based on subjective assessment of lameness, but no measurable treatment effects on objective measures of limb lameness. Within the axial skeleton there were significant treatment effects on pain in the polo horses and stiffness in the Quarter Horses. No significant differences were noted within the active range of motion (i.e., baited stretches) or induced spinal reflexes.

### 4.1. Subjects

The subjects used in this study were all clinical cases with varying degrees of lameness and axial skeleton pain and dysfunction, which contributed to the increased variability in baseline values and likely affected the reported responses to treatment. It is difficult to collect a uniform sample of subjects with similar types or degrees of limb lameness or axial skeleton pain from routine clinical cases, which is a good, but costly justification, for using experimental models for inducing lameness [44,45] or back pain [46,47].

For most outcome measures there were significant differences detected between the polo and Quarter Horses, which we judged to be due to the management (i.e., collegiate versus privately owned) and the demands of ridden exercise or competition. Due to the perceived high variability in outcome measures in the polo horses, we attempted to limit variability by enrolling a more uniform sample of Quarter Horses with localized proximal hind limb lameness, which did produce significant differences in measures of axial skeleton stiffness and MNT values but did not consistently affect any reported lameness parameters.

Unfortunately, the polo horses began their competition season at the beginning of the study timeline. The polo horses were rested from summer turnout and experienced a sudden increase in ridden exercise during the study timeline, which likely negatively influenced (i.e., increased subjective hind limb and whole-body lameness scores at week 4) and increased variability in measures of lameness and concurrent neck and back pain. These sources of error were addressed in the Quarter Horses with the inclusion of privately owned, consistently ridden horses that did not experience any changes in workload during the study timeline.

There was a wide age range reported in both polo (range 5 to 23) and Quarter Horses (range 6 to 22) horses. While all horses were working or actively competing it is possible that some of the older horses did have more functional limitations and accumulated tissue degeneration, which could have negatively impacted some of the measured outcome parameters (e.g., active range of motion, spinal reflexes, and compensatory lameness mechanisms) [48].

### 4.2. Subjective Lameness Evaluation

Many of the polo horses had chronic, multiple limb lameness; some of which had persisted for long periods of time with minimal treatment. It was expected that these horses had deeply ingrained compensatory lameness mechanisms [49] and concurrent axial skeleton pain and dysfunction due to strenuous athletic use during the competition season and inconsistent levels of equestrian skills among the collegiate riders [50,51]. This was supported by the higher baseline whole-body lameness scores in the polo horses (range 4.3 to 4.5), compared to the Quarter Horses (range 2.8 to 3.6).

The chronicity of lameness and the increased likelihood for the development of peripheral and central sensitization (i.e., neuropathic pain) may have also contributed to the limb lameness being refractory to conservative treatment [52,53]. The privately owned Quarter Horses were selected with the intent of creating a more consistent sample of horses with isolated hind limb lameness as evidenced by higher baseline hind limb lameness scores (range 4.1 to 4.9), compared to the polo horses (range 2.5 to 2.7). Measured differences between the polo and Quarter Horses was likely due to different sample populations and subjective lameness evaluation by two separate examiners.

### 4.3. Objective Lameness Evaluation

All objective measures of forelimb lameness had relatively large variation, which carried over into the whole-body measures of lameness and likely impacted the statistical results. Repeat attempts at inertial sensor data collection were needed for most horses based on required gait parameters (e.g., number of acceptable strides, the magnitude and standard deviation of the vector sum).

The inertial sensor system used in the study is optimally designed to identify single limb lameness and has limitations in quantifying multiple limb lameness [54]. Most inertial sensor-based studies focus on capturing measures of limb lameness with little to no regard for the potential confounding effects of axial skeleton pain or dysfunction [55,56]. Typical inertial sensor placement is on the dorsum of the head and pelvis [35]; therefore, it is expected that altered head and pelvic displacement due to axial skeleton pain or stiffness would directly affect measures of limb lameness [57,58]. The large variability in the reported vector sum values for the forelimbs supports this. Altered spinal kinematics (i.e., joint range of motion) has been reported in horses with natural occurring and experimentally induced back pain [46,59]; However, changes in vertical displacement or acceleration of the poll, withers, or pelvis as measured with inertial sensors have not been reported in horses with neck, back or pelvic pain to date [60,61,62]. It is theorized that neck pain and stiffness would alter the vertical acceleration of the head and lumbosacral pain and stiffness would affect the vertical acceleration of the pelvis. Further research is needed to investigate the effect and magnitude of axial skeleton pain and stiffness on inertial parameters used to identify the presence, localization, and severity of limb lameness as most horses with chronic limb lameness also have varying degrees of concurrent neck or back pain and stiffness [12,13,14,15].

### 4.4. Spinal Evaluation

Clinical experience suggests that forelimb lameness is more closely associated with cervical pain and dysfunction and that horses with back pain may have concurrent hind limb lameness issues (and vice versa) [56,63,64]. Therefore, we expected to observe higher cervical pain and dysfunction scores in the polo horses (i.e., primary forelimb lameness) and more prevalent thoracolumbar issues in the Quarter Horses selected for the presence of hind limb lameness. Surprisingly, the polo horses had much higher baseline pain scores within the lumbopelvic region (range 6.0 to 8.9), compared to the Quarter Horses (range 0.9 to 1.1). We theorized that the polo horses would have more compensatory spinal issues due to being ridden by different collegiate riders of varying skill levels; however, baseline values for stiffness and muscle hypertonicity were consistently lower across all vertebral regions, compared to the Quarter Horses.

A significant contributing factor to measured differences between years was likely due to spinal evaluation by two different examiners. There is evidence that inter-examiner reliability of manual examination techniques by physiotherapists is repeatable [65]. The same parameters for examination were used in both years, and the guidelines to interpret pain responses to palpation were based on previous established criteria to evaluate for pain, hypertonicity, and stiffness in horses [36,66]. Despite the difference in examiners between years, significant treatment effects were noted in measures of whole-body pain severity scores in the polo horses and number of affected vertebrae in whole-body stiffness scores in the Quarter Horses. Little to no treatment effects were noted on measures of muscle hypertonicity as reported previously in horses treated with acute back pain [38]. However, other chiropractic studies that have shown positive treatment effects in horses with chronic back pain and muscle hypertonicity [22,67,68]. Measuring treatment effects immediately post-treatment typically produces significant changes [69]; whereas, measuring treatment effects a week after the applied treatment often produces varying results [43]. As our goal was to assess global changes over time, we choose to only collect outcome measures prior to treatment and not immediately pre- and post-treatment.

### 4.5. Active Range of Motion

Reduced stiffness (i.e., improved spinal mobility) as assessed with passive joint mobilization would be expected to be associated with a concurrent increase in active range of motion induced during the baited stretches. While the active range of motion scores visibly improved over time in both the polo and Quarter Horses, the changes were not statistically significant. Prior studies have shown improvements in passive trunk mobility following chiropractic care [42,43] and when combined with low-level laser therapy [38]. Active cervical range of motion has been used as an outcome parameter to assess the therapeutic response to a deep tissue heating modality; however, no significant treatment effects were reported [70]. Active range of motion (i.e., baited stretches) has been used in numerous studies to assess changes in thoracolumbar multifidi muscle cross-sectional area [39,71]. However, active range of motion techniques have not yet been validated for use as an outcome parameter for assessing neck or back stiffness in horses.

### 4.6. Spinal Reflexes

Spinal reflexes were included as potential measures of neuromuscular coordination and motor control [38]. While the cumulative spinal reflex scores decreased (i.e., improved) over time in both the polo and Quarter Horses, the changes were not statistically significant. Again, spinal reflexes have been used extensively in the clinical setting but have not yet been validated as an outcome parameter in a research setting for horses [70].

### 4.7. Mechanical Nociceptive Thresholds

There were mixed results in the MNT values in both treatment and control groups. Treatment effects included significant MNT increases (i.e., less pain) within the lumbosacral region in the Quarter Horses with primary hind limb lameness. Prior equine chiropractic studies that have included MNTs as an outcome parameter have reported larger and more consistent positive treatment responses in sound horses with back pain [22,67]. Significant MNT decreases (i.e., more pain) within the cervical region in the polo horses may have reflected inconsistent riders, initiation of active competition, and increased pulling on the head and neck. Concurrent limb lameness and potential central sensitization may limit treatment effects within the axial skeleton in affected horses. However, spinal mobilization in a rodent model has been reported to reduce peripheral sensitization within the limbs [72].

### 4.8. Chiropractic Treatment

While chiropractic treatment is typically applied with the intent of producing positive or beneficial effects, there is evidence that spinal manipulation in horses may have short-term negative effects (i.e., more painful 1 day post treatment) [73], or is less effective in horses with acute back pain [38]. In the current study, all outcome measures were collected prior to any applied treatment at 2 and 4 weeks, which should have minimized immediate post-treatment effects. All horses were judged to have chronic musculoskeletal disorders; however, acute bouts of inflammation could have hampered measured treatment effects.

Chiropractic treatment was applied once a week over four weeks. This treatment frequency has shown positive effects in prior equine chiropractic studies [43,67]. While the experimental design used in this study may not fully reflect the clinical setting, we had to balance the demands of providing a perceived effective treatment with completing the research in a timely manner on a large number of client-owned horses. It is possible that a different frequency or duration of chiropractic treatment may have produced more favorable results.

### 4.9. Limitations

The outcome measures used in this study were carefully considered to provide a global measure of limb lameness and concurrent axial skeletal pain and dysfunction. However, several of the included parameters were judged to be subjective and have not been validated using established pain models [46]. Unfortunately, the past medical history and duration of limb lameness was inconsistent or not available for most horses. A complete medical history may have helped us to better focus our treatment on specific body regions or to avoid treatment at some affected articulations. We did not perform a comprehensive diagnostic evaluation (i.e., diagnostic local anesthesia, diagnostic imaging) needed to localize and confirm the source of limb lameness or axial skeleton pain and dysfunction within individual horses. Reports of forelimb lameness associated with cervical disease and hind limb lameness producing signs of back pain often contribute to the diagnostic challenge in determining the primary source of pain or lameness in affected horses and subsequent treatment approaches [56,63,64]. Confirming a definitive diagnosis is difficult when there is a partial or incomplete response to repeated or progressive diagnostic local anesthesia from the distal to proximal aspects of the affected fore or hind limbs. Establishing clinical relevance is also difficult if multiple radiographic lesions are identified within both axial and appendicular skeleton locations [74]. Therefore, we selected functional outcome parameters that were judged to be representative of the whole-body effects of pain and dysfunction, irrespective of the perceived source or location [38]. Poor correlations have been reported between performance [75,76], clinical signs [5], and radiographic findings [6] and their inclusion would likely not have offered a measurable benefit in the horses with chronic, multilimb lameness as the precipitating source of pain or lameness was not likely limited to a single site, especially in the presence of peripheral or central sensitization [53]. The use of different examiners between years likely contributed to differences between the polo and Quarter Horses, but the magnitude of this effect on the results is unknown.

### 4.10. Future Directions

Future chiropractic-lameness studies need to limit enrollment to horses with a single limb, subacute lameness of known origin to help reduce the variability associated with compensatory, multilimb lameness. Validated outcome parameters that capture functional impairments and disability in horses with axial skeleton pain and dysfunction is critically needed [77,78,79]. A deeper understanding of the pain pathophysiology associated with fore and hind limb lameness and the development of the compensatory gait patterns associated with the axial skeleton would provide useful insights into how chiropractic care may be tailored to optimize adjunctive support for horses with chronic limb lameness. A whole-body approach for investigating lameness and related axial skeleton issues during athletic activities is becoming more of a reality with the development of advanced inertial sensor systems; however, the effect of neck and back pain or stiffness on inertial placement and parameters used to identify limb lameness is unknown [57,60,80].

## 5. Conclusions

In conclusion, our results suggest that chiropractic treatment has mixed therapeutic effects on measures of lameness and axial skeleton pain and dysfunction as applied in this study. Positive treatment effects were noted in subjective measures of hind limb and whole-body lameness, back pain and stiffness, and MNT values within the lumbosacral region of horses with primary hind limb lameness. Limitations of the study included the lack of pathoanatomical diagnoses, multilimb lameness, inter-examiner variability, and lack of validated outcome measures. Further studies are needed to better identify the type and severity of limb lameness that may be amendable to chiropractic treatment.

## Figures and Tables

**Figure 1 animals-12-02845-f001:**
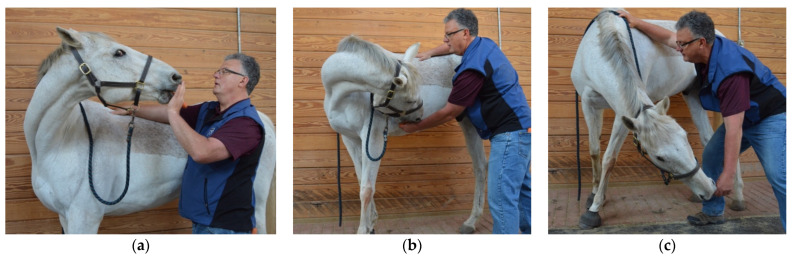
Baited stretches used to assess the range of motion and fluidity of active lateral bending of the (**a**) cranial; (**b**) middle; and (**c**) caudal cervical regions.

**Figure 2 animals-12-02845-f002:**
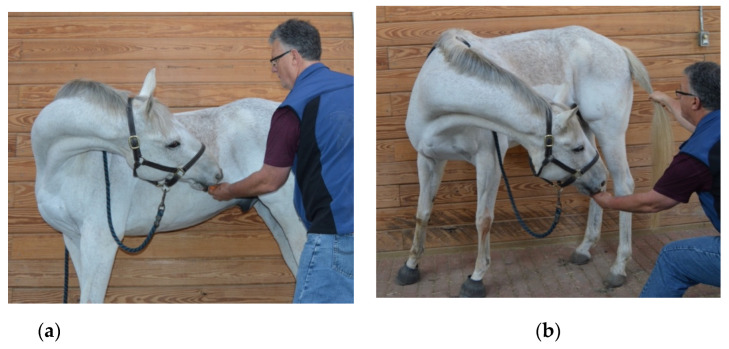
Baited stretches used to assess the range of motion and fluidity of active lateral bending of the combined cervical and thoracolumbar vertebral regions with a target directed toward the (**a**) lateral stifle; and (**b**) lateral tarsal regions. Note the increased activation and flexion of the trunk in (**b**), compared to the neutral trunk posture and stance in (**a**).

**Figure 3 animals-12-02845-f003:**
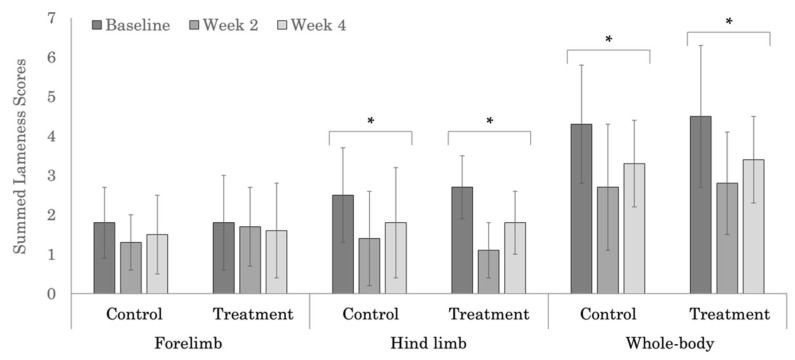
Summed forelimb, hind limb, and whole-body subjective lameness scores for the polo horses. * Indicates significant differences across weeks, within groups (*p* < 0.05).

**Figure 4 animals-12-02845-f004:**
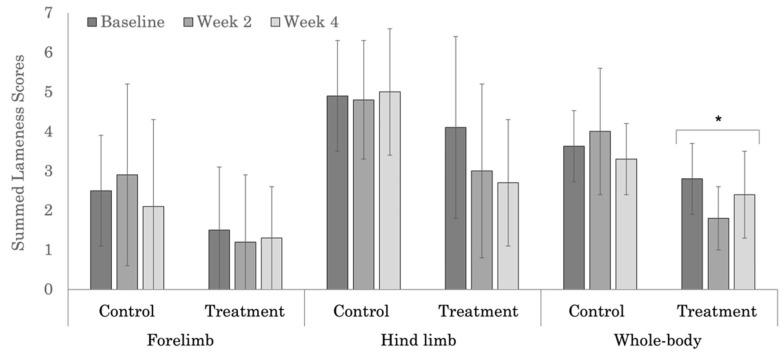
Summed forelimb, hind limb, and whole-body subjective lameness scores for the Quarter Horses. * Indicates significant differences across weeks, within groups (*p* < 0.05).

**Figure 5 animals-12-02845-f005:**
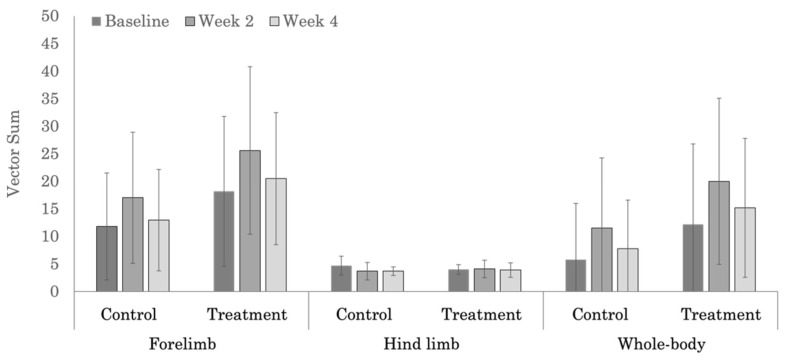
Vector sum values used for assessing the severity of forelimb, hind limb, and whole-body lameness in the polo horses.

**Figure 6 animals-12-02845-f006:**
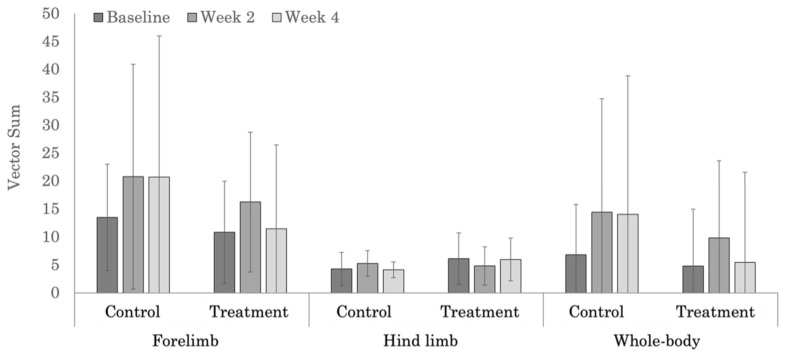
Vector sum values used for assessing the severity of forelimb, hind limb, and whole-body lameness in the Quarter Horses.

**Figure 7 animals-12-02845-f007:**
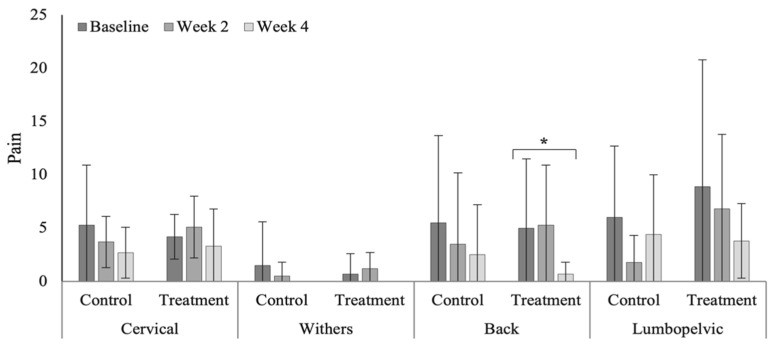
Pain scores across vertebral regions in the polo horses. * Indicates significant differences across weeks, within groups (*p* < 0.05).

**Figure 8 animals-12-02845-f008:**
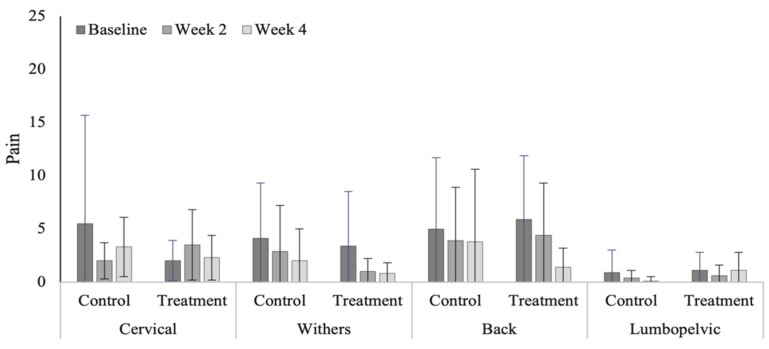
Pain scores across vertebral regions in the Quarter Horses.

**Figure 9 animals-12-02845-f009:**
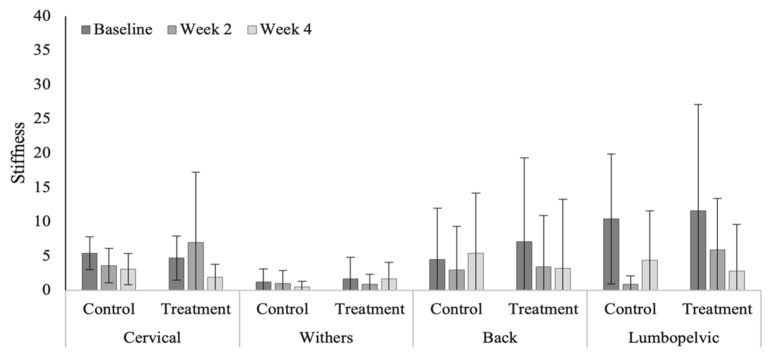
Stiffness scores across vertebral regions in the polo horses.

**Figure 10 animals-12-02845-f010:**
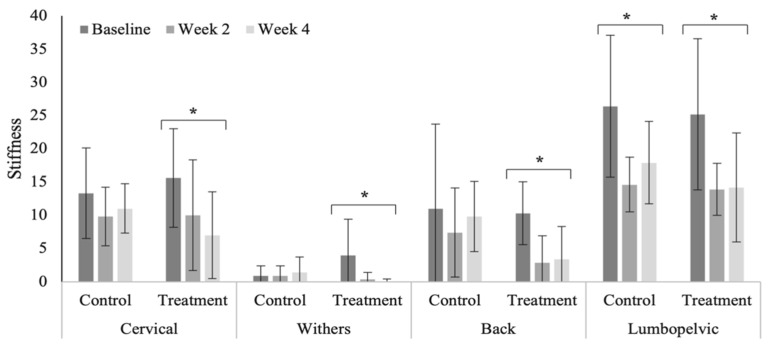
Stiffness scores across vertebral regions in the Quarter Horses. * Indicates significant differences across weeks, within groups (*p* < 0.05).

**Figure 11 animals-12-02845-f011:**
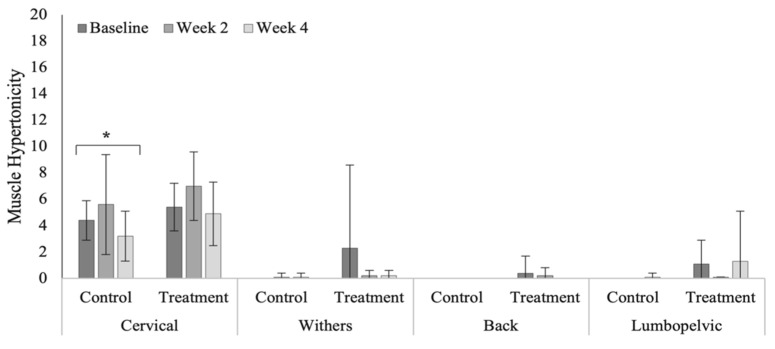
Muscle hypertonicity across vertebral regions in the polo horses. * Indicates significant differences across weeks, within groups (*p* < 0.05).

**Figure 12 animals-12-02845-f012:**
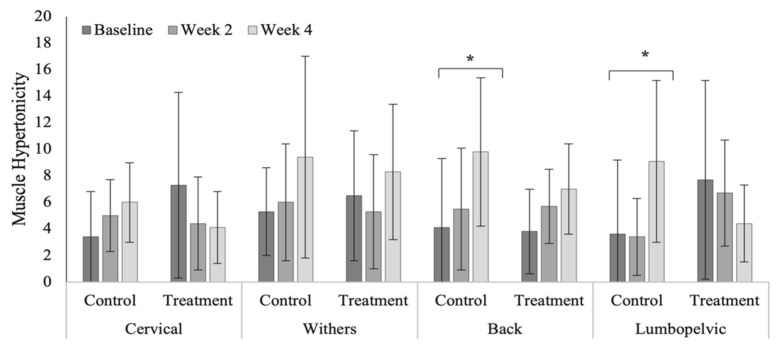
Muscle hypertonicity across vertebral regions in the Quarter Horses. * Indicates significant differences across weeks, within groups (*p* < 0.05).

**Figure 13 animals-12-02845-f013:**
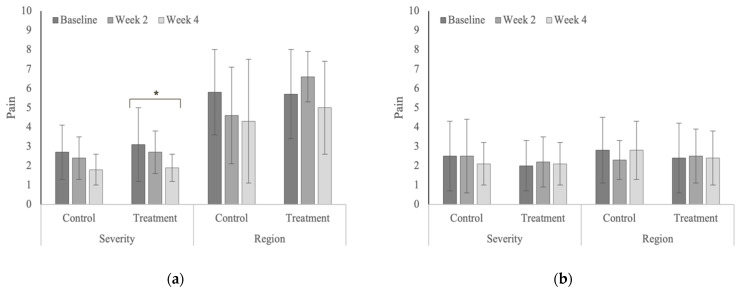
Whole-body scores for the severity and affected regions of pain (**a**) in the polo horses; (**b**) in the Quarter Horses. * Indicates significant differences across weeks, within groups (*p* < 0.05).

**Figure 14 animals-12-02845-f014:**
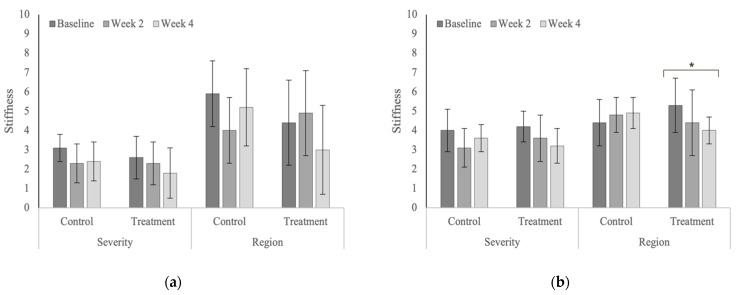
Whole-body scores for the severity and affected regions of stiffness (**a**) in the polo horses; (**b**) in the Quarter Horses. * Indicates significant differences across weeks, within groups (*p* < 0.05).

**Figure 15 animals-12-02845-f015:**
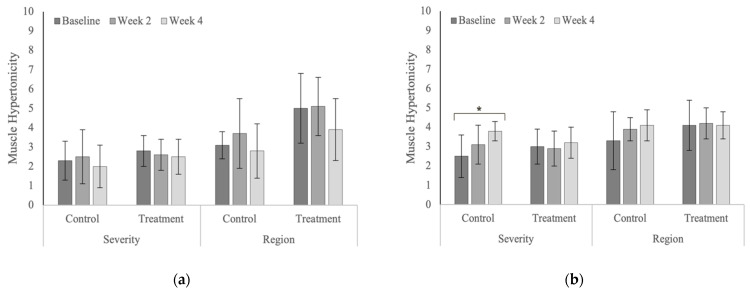
Whole-body scores for the severity and affected regions of muscle hypertonicity (**a**) in the polo horses; (**b**) in the Quarter Horses. * Indicates significant differences across weeks, within groups (*p* < 0.05).

**Figure 16 animals-12-02845-f016:**
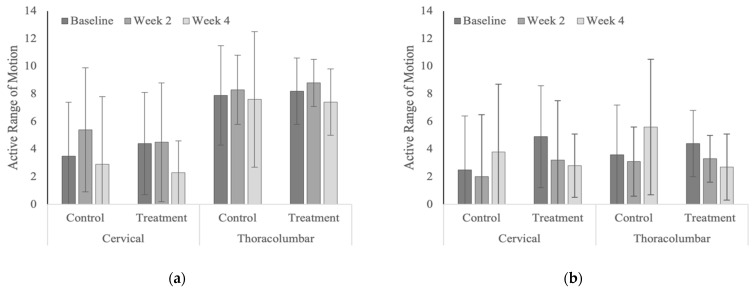
Summed active range of motion scores within vertebral regions (**a**) in the polo horses; (**b**) in the Quarter Horses.

**Figure 17 animals-12-02845-f017:**
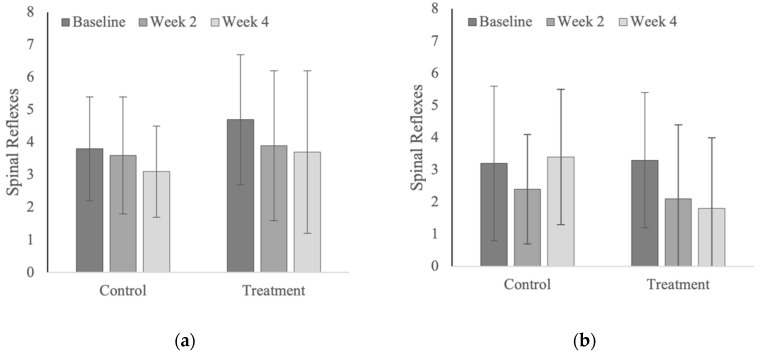
Summed spinal reflex scores within vertebral regions (**a**) in the polo horses; (**b**) in the Quarter Horses.

**Figure 18 animals-12-02845-f018:**
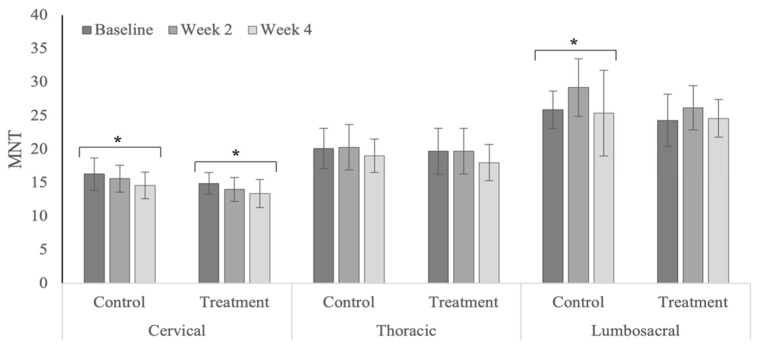
Mechanical nociceptive threshold values within spinal regions in the polo horses. * Indicates significant differences across weeks, within groups (*p* < 0.05).

**Figure 19 animals-12-02845-f019:**
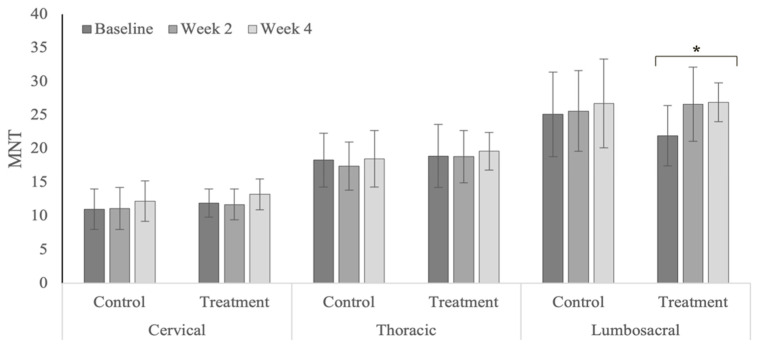
Mechanical nociceptive threshold values within spinal regions in the Quarter Horses. * Indicates significant differences across weeks, within groups (*p* < 0.05).

## Data Availability

Data available upon request from the corresponding author (K.K.H.).

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
