# Peer review of "The Effect of Chiropractic Treatment on Limb Lameness and Concurrent Axial Skeleton Pain and Dysfunction in Horses"

_animals, 2022, doi:10.3390/ani12202845_

Round 1

Reviewer 1 Report

Dear Authors

The article submitted for review is interesting. I recommend minor revisions to the manuscript before possible publication:
1. Please complete the Conclusions section; as it stands, it is unsatisfactorily written.
2. Please follow the guidelines for Authors on how to cite articles in Animals journal throughout the manuscript.

Author Response

Please complete the Conclusions section; as it stands, it is unsatisfactorily written.

  • Line 623 - Conclusions were revised

Please follow the guidelines for Authors on how to cite articles in Animals journal throughout the manuscript.

  • Periods were moved from before to after the citations.

Reviewer 2 Report

General concept comments: 

There is great merit in what you are trying to achieve with this paper, and it is a topic that is highly relevant to current veterinary medicine and equestrianism. I think there are lots of very strong points including the design of the trial which includes subjective and objective measures of lameness and pain. 

The age range of horses, whilst interesting to consider in more detail in the future, is very wide. I think age could play a role in outcomes in this paper and it might be worth mentioning this in your discussion. Older horses may be less able to perform some of the ROM/baited stretches or rounding as well as having a different ability to recover from lameness of certain types or muscle deficiencies.

Only in year 1 forelimb treatment group and hindlimb year 2 treatment groups did summed subjective lameness score not rebound in week 4. Further to this, I think it is worth mentioning that in the objective measurement of lameness, in year 1 and year 2, treatment group vector sum value actually increased in week 2 in the forelimbs. It also slightly increased in year 1 week 2 for the hindlimbs and increased in the whole body lameness group. We need to be aware of possible negative effects as well as positive effects of chiropractic treatment. Whilst you cannot conclude this effect is due to the treatment, it must be considered that there could be adverse effects of chiropractic treatment in some cases. 

It is a limitation forelimb lameness caused by cervical disease was not determined in this paper. If cervical pain was the cause of forelimb lameness, then this could alter a horse’s response to chiropractic treatments including ROM exercises if you compared that to a lameness coming from the foot for example. Even simply dividing horses into pain caused by axial versus appendicular skeletal lameness could have changed the grouping outcomes and perhaps provided a different outcome. However I appreciate that the aim here was a “whole body” approach. I think clinically this differentiation is very important when recommending treatment types for our patients particularly in reference to forelimb lameness caused by cervical spinal disease.

Specific comments: 

Line 15: consider alternative word to “battery”. 

Line 18: is this muscular or bony back pain or both? 

Line 26: stiffness of what? Lack of flexibility of the spine? Or stiffness of the legs? 

Line 39-49: It is a large generalization to suggest that diagnostic imaging as a whole does not correlate well with diagnostic analgesia based on one paper about small tarsal joint osteoarthritis. There are many other conditions where diagnostic imaging (which includes advanced imaging such as CT and MRI) correlates well with diagnostic analgesia. However it is fair to say that at times, diagnostic imaging may not always provide a definitive diagnosis despite localisation of lameness with diagnostic analgesia. 

Line 52: Current terminology for pain and degeneration of the sacroiliac region would be sacroiliac dysfunction or disorder as it is not just a process of osteoarthritis but includes ligamentous injury, possible muscular dysfunction etc. 

Line 62-64: Non-pharmaceutical approaches are not necessarily “required” at FEI sanctioned events, and some non-pharmaceutical approaches cannot be carried out at FEI events. You could instead suggest that non-pharmaceutical approaches such as chiropractic practices could be useful at FEI events and may be preferred by owners for these reasons (as you list above). 

Line 65: Opinion has crept into this line where “well-placed” is used. Reword this as this is a clinical trial not a review or opinion piece. I understand your point, however your final line of this paragraph says what you are trying to get across already without opinion. 

Line 74-76: The referenced review stated it was difficult to draw strong conclusions about musculoskeletal mobilization despite reported positive effects, so I think it is unfair to suggest there is a “high level of efficacy” for these techniques based on this paper.

Line 89: Reword this as “within at least one fore or hindlimb were evaluated” does not read well. 

Line 98-99: The use of flexion tests to rule in or out the hock and stifle alone is not definitive. All flexion tests performed in this region also flex the coxofemoral joint. However given your hypothesis is “irrespective of the perceived sites or sources of pain” this is probably not important in your outcomes. 

Line 122: Was the surface hard or soft for the subjective lameness evaluation? Were both surfaces examined. 

Line 124: were referred lameness cases accounted for? Or was any form of lameness counted as part of the score. Some grade 3/5 hindlimb lameness cases will have a referred forelimb lameness that is not a primary lameness and vice versa.

Line 168: Change to “complete” not compete. 

Line 389: Summed spinal reflex score also tended to improve in the control group in year 1 not just the treatment group. 

Line 396: Change to “i.e. more painful” not more pain. 

Line 539: Why would this have not provided measurable benefit? Benefit to what? I assume you mean benefit to your hypothesis and outcomes, which you should spell out. 

Table A4, A5, A6: The title needs to stand alone with no further reading required. You should include something like, vector sum values ( i.e. the total head height difference indicating forelimb lameness where >8.5 is considered to be a forelimb lameness). 

Author Response

General concept comments: 

There is great merit in what you are trying to achieve with this paper, and it is a topic that is highly relevant to current veterinary medicine and equestrianism. I think there are lots of very strong points including the design of the trial which includes subjective and objective measures of lameness and pain.

The age range of horses, whilst interesting to consider in more detail in the future, is very wide. I think age could play a role in outcomes in this paper and it might be worth mentioning this in your discussion. Older horses may be less able to perform some of the ROM/baited stretches or rounding as well as having a different ability to recover from lameness of certain types or muscle deficiencies.

  • Line 452 - Added sentence on wide age ranges and reference on age effects
  • We did complete statistical analysis on age effects and the results were mixed. In an effort to keep things simple and to not further lengthen the manuscript, we chose to not include this data.  If the reviewer feels strongly about its inclusion, we can comply.
    • There were no significant age differences between the polo and Quarter Horses (p = 0.061)

Only in year 1 forelimb treatment group and hindlimb year 2 treatment groups did summed subjective lameness score not rebound in week 4.

  • Line 447 - Added comment to discussion

Further to this, I think it is worth mentioning that in the objective measurement of lameness, in year 1 and year 2, treatment group vector sum value actually increased in week 2 in the forelimbs. It also slightly increased in year 1 week 2 for the hindlimbs and increased in the whole body lameness group. We need to be aware of possible negative effects as well as positive effects of chiropractic treatment. Whilst you cannot conclude this effect is due to the treatment, it must be considered that there could be adverse effects of chiropractic treatment in some cases.

  • Line 565 - Added discussion on potential adverse effects

It is a limitation forelimb lameness caused by cervical disease was not determined in this paper. If cervical pain was the cause of forelimb lameness, then this could alter a horse’s response to chiropractic treatments including ROM exercises if you compared that to a lameness coming from the foot for example. Even simply dividing horses into pain caused by axial versus appendicular skeletal lameness could have changed the grouping outcomes and perhaps provided a different outcome. However I appreciate that the aim here was a “whole body” approach. I think clinically this differentiation is very important when recommending treatment types for our patients particularly in reference to forelimb lameness caused by cervical spinal disease.

  • Line 591 - Added discussion on forelimb lameness and cervical disease

Reviewer 3 Report

This study was carefully designed, well conducted, and despite the great statistical ambiguity it lead to important conclusions regarding chiropractic treatment in horses. This is all the more impressive considering that the study was borne out of opportunity, and features two test populations with barely controllable medical backgrounds. The writing is clear and concise, and features few grammatical and typographic errors.

Below I request a few clarifications for study design and discussion.

In addition, I wonder if there is a way to show differences in individuals, instead of examining entire treatment groups.  There seems to be a treatment effect, but it is too small to be significant for the whole sample. Are other biometrics available (ANOVA with age, sex, body height, or mass as predictor) that could be used to discern which patients reacted positively to the treatment and which didn’t?

88: This verbal distinction between “year 1” and “year 2” bothers me throughout the manuscript, because it implies that the study year makes a difference. However, from what I understand the study populations are vastly different between these two samples, and I think it would be more tangible to call them “Polo” and “Show” horses, or similar names that indicate the differential origin of these samples.

98: Why were forelimb-lame patients excluded from the second group, but not from the first one? This is mentioned in the Discussion, but should be indicated here.

209: Descriptions and images of the baited stretches are excellent.

260: Why were the patients treated with this relatively low frequency? Applying a 15 minute treatment every seven days seems relatively rare, compared to potentially daily exercise.

485: Are the Year 1 horses bred for polo? One might suspect that particular breeding lines are more resilient to vertebral injury and feature compensatory movements that diverge from “normal” horses.

497: Please reflect on why those studies might have found a positive effect where you found none. Were the study designs markedly different? Was therapy provided at greater volume or frequency? Or were the positive effects in previous studies small enough to be overlooked as random effects in your study?

504: Refer to the associated Figure and Table directly. The effect is not statistically significant, but is still visible.

550: I suspect that the results would be clearer if horses in the study had been exempt from performance or competition for the duration of the study. Differential treatment of the patients during those competitions and shows potentially added a strong random effect to all data.

Author Response

Below I request a few clarifications for study design and discussion.

In addition, I wonder if there is a way to show differences in individuals, instead of examining entire treatment groups.

  • The individual data can be made available via supplemental material to those interested parties. However, we judged the current large number of figures and included tables in the appendix to be useful without being excessive for publication purposes.  Adding individual data would be very unwieldy and undermines the purpose of reporting statical significance, unless there are clear trends in individual horses, which we did not observe.
  • We did replace all tables in the text with graphs in any effort to improve readability. All mentioned tables were moved to the appendix.

There seems to be a treatment effect, but it is too small to be significant for the whole sample. Are other biometrics available (ANOVA with age, sex, body height, or mass as predictor) that could be used to discern which patients reacted positively to the treatment and which didn’t?

  • Line 453 – Added text about age effects on lameness. We did not collect any wither height or body weight data on the horses.
  • There were minimal significant age and no gender effects observed across outcome parameters. The only significant age effect was noted for stiffness in both groups (treatment; p=0.016 and control p=0.035).

88: This verbal distinction between “year 1” and “year 2” bothers me throughout the manuscript, because it implies that the study year makes a difference. However, from what I understand the study populations are vastly different between these two samples, and I think it would be more tangible to call them “Polo” and “Show” horses, or similar names that indicate the differential origin of these samples.

  • Line 436 - The sample populations were different between years across most of the measured variables.
  • The labels for year 1 and year 2 were changed to polo and Quarter Horses throughout the text.

98: Why were forelimb-lame patients excluded from the second group, but not from the first one? This is mentioned in the Discussion but should be indicated here.

  • Line 87 and 438 - Added text.

209: Descriptions and images of the baited stretches are excellent.

  • Thank you for the compliment as we strived to make the extensive methods as clear as possible.

260: Why were the patients treated with this relatively low frequency? Applying a 15 minute treatment every seven days seems relatively rare, compared to potentially daily exercise.

  • Line 569 - Serial treatments applied every 7 days over several sessions has been used in prior equine chiropractic studies. While the experimental design may not fully reflect the clinical setting, we had to balance the demands of providing a perceived effective treatment with completing the research in a timely manner on a large number of client-owned horses.
  • The treatment effect of daily exercise has not been evaluated in horses with lameness or axial skeleton pain and dysfunction. While studies on the effects of exercise and comparative studies assessing the efficacy of exercise and spinal mobilization or manipulation have been done in humans with low back pain, the authors are unaware of similar research or knowledge in horses.

485: Are the Year 1 horses bred for polo? One might suspect that particular breeding lines are more resilient to vertebral injury and feature compensatory movements that diverge from “normal” horses.

  • Line 84 - The polo horses were a population of mixed breeds that were donated to the college and not purpose bred.

497: Please reflect on why those studies might have found a positive effect where you found none. Were the study designs markedly different? Was therapy provided at greater volume or frequency? Or were the positive effects in previous studies small enough to be overlooked as random effects in your study?

  • Line 524 – Discussion was added.

504: Refer to the associated Figure and Table directly. The effect is not statistically significant, but is still visible.

  • Line 534 – It is not customary to call out specific figures or tables in the discussion. However, the authors are willing to add references to specific figures and tables throughout the discussion if the reviewer feels strongly about this.

550: I suspect that the results would be clearer if horses in the study had been exempt from performance or competition for the duration of the study. Differential treatment of the patients during those competitions and shows potentially added a strong random effect to all data.

  • Line 436 - The authors agree that treatment effects might have been less confounded without the added demands of ridden exercise or changes in exercise intensity. Despite all our best made plans, there were other factors such as changes in ridden exercise that were beyond our control.

Specific comments: 

Line 15: consider alternative word to “battery”. 

  • Line 14 - Changed word

Line 18: is this muscular or bony back pain or both?

  • Line 17 – Added text

Line 26: stiffness of what? Lack of flexibility of the spine? Or stiffness of the legs?

  • Line 25 – Added text

Line 39-49: It is a large generalization to suggest that diagnostic imaging as a whole does not correlate well with diagnostic analgesia based on one paper about small tarsal joint osteoarthritis. There are many other conditions where diagnostic imaging (which includes advanced imaging such as CT and MRI) correlates well with diagnostic analgesia. However it is fair to say that at times, diagnostic imaging may not always provide a definitive diagnosis despite localization of lameness with diagnostic analgesia.

  • Line 44 – The authors agree that we overgeneralized our initial comments. Added text and supporting references.

Line 52: Current terminology for pain and degeneration of the sacroiliac region would be sacroiliac dysfunction or disorder as it is not just a process of osteoarthritis but includes ligamentous injury, possible muscular dysfunction etc.

  • Line 52 - Changed terms

Line 62-64: Non-pharmaceutical approaches are not necessarily “required” at FEI sanctioned events, and some non-pharmaceutical approaches cannot be carried out at FEI events. You could instead suggest that non-pharmaceutical approaches such as chiropractic practices could be useful at FEI events and may be preferred by owners for these reasons (as you list above).

  • Line 63 – In an effort to simplify, deleted comment about FEI.

Line 65: Opinion has crept into this line where “well-placed” is used. Reword this as this is a clinical trial not a review or opinion piece. I understand your point, however your final line of this paragraph says what you are trying to get across already without opinion.

  • Line 63 – Removed “well-placed” and reworded sentence. The statement reflects results in the mentioned references [22, 23].

Line 74-76: The referenced review stated it was difficult to draw strong conclusions about musculoskeletal mobilization despite reported positive effects, so I think it is unfair to suggest there is a “high level of efficacy” for these techniques based on this paper.

  • Line 71 – Removed ‘mobilization’ but the authors stand by the comment of a high level of efficacy for spinal manipulation for the listed outcome parameters.

Line 89: Reword this as “within at least one fore or hindlimb were evaluated” does not read well.

  • Line 86 - Reworded sentence

Line 98-99: The use of flexion tests to rule in or out the hock and stifle alone is not definitive. All flexion tests performed in this region also flex the coxofemoral joint. However given your hypothesis is “irrespective of the perceived sites or sources of pain” this is probably not important in your outcomes.

  • Line 97 - Reworded sentence

Line 122: Was the surface hard or soft for the subjective lameness evaluation? Were both surfaces examined.

  • Line 121 – Added text

Line 124: were referred lameness cases accounted for? Or was any form of lameness counted as part of the score. Some grade 3/5 hindlimb lameness cases will have a referred forelimb lameness that is not a primary lameness and vice versa.

  • Line 122 – Added sentence

Line 168: Change to “complete” not compete.

  • Line 168 - Word changed

Line 389: Summed spinal reflex score also tended to improve in the control group in year 1 not just the treatment group.

  • Line 403 - Added text

Line 396: Change to “i.e. more painful” not more pain.

  • Lines 408 and 410 - Corrected terms

Line 539: Why would this have not provided measurable benefit? Benefit to what? I assume you mean benefit to your hypothesis and outcomes, which you should spell out.

  • Line 603 – Added supporting text

Table A4, A5, A6: The title needs to stand alone with no further reading required. You should include something like, vector sum values (i.e. the total head height difference indicating forelimb lameness where >8.5 is considered to be a forelimb lameness).

  • The authors agree in principle that all figure and table titles should be able to stand alone. However, very few of the included figure or table titles are truly able to stand alone as the methodology used for many of the outcome parameters are not standardized nor widely accepted.  Therefore, the reader must rely on the detailed descriptions within the materials and methods to gain a full understanding of the terms used in the titles.
  • The author guidelines suggest the use of simple, concise descriptions for figures and tables, which the authors believe we have done. However, we are open to expanding the titles if the reviewer feels strongly about this issue.